# Sustainable Solutions for Oyster Shell Waste Recycling in Thailand and the Philippines

**Ramakrishna Chilakala [1], Chottitisupawong Thannaree [2], Eunsoo Justin Shin [3], Thriveni Thenepalli [4] and Ji Whan Ahn [4],***

[1] Department of Bio-based Materials, School of Agriculture and Life Science, Chungnam National University, Daejeon City 34134, Korea

[2] Department of Chemical and Biomolecular Engineering, Korea Advanced Institute of Science and Technology, 291 Daehak-ro, Yuseong-gu, Daejeon 34141, Korea

[3] International School Manila (ISM), University Parkway, Fort Bonifacio Global City, Taguig City 1634, Philippines

[4] Center for Carbon Mineralization, Mineral Resources Division, Korea Institute of Geosciences and Mineral Resources (KIGAM), 124 Gwahagno, Gajeong-dong, Yuseong-gu, Daejeon 34132, Korea

* Correspondence: ahnjw@kigam.re.kr; Tel.: +82-042-868-3578; Fax: +82-042-861-3990

**Abstract:** This paper studies the utilization and management of the waste mollusk shell. The two major export countries of mollusk shell are the Southeast Asia's Thailand and the Philippines. First, the aquaculture of oysters and bivalve shells has been studied as background understanding. The effect of the global climate change on farming and the consequences of farming on the nearby environment and neighborhoods have also been discussed. The utilization technologies on the waste shell are available on a small scale and not industrialized. This study offers an enabling context under which a suitable method can take action to solve the overflow waste shell problem, and at the same time, provide sustainable management.

**Keywords:** oyster shell waste; Thailand; Philippines; environment; climate change; sustainable management

## 1. Introduction

The oyster is a type of bivalve mollusk that grows in brisk water. It can be naturally found in the coastal area where the fresh water meets brine water. Similar to many other living organisms, oysters require some specific environmental conditions to survive, but in general, oysters can be found in most of the world. Oysters are known for being well balanced in maintaing their rich nutrition value high in protein and minerals, while also maintaining low fat, calories, and cholesterol. Due to their nutrition value, oysters are the top choice for many consumers. For hundreds of years, the oyster has been a common food in the coastal areas around the globe. Due to the increase in demand, the aquaculture technologies were developed to meet the requirements. Aquaculture is economically improving, but on the environmental side, it is still uncertain. Overcultivation without taking account of the social costs would constrain the stability of the coastal environment. One of the upcoming problems is the waste shell from the oyster. In Southeast Asia, particularly Thailand and the Philippines, two countries with the highest oyster aquaculture production, the waste shell recycling methods are not developed properly and the shells are mostly dumped as a part of food waste. Improper management of the shells tends to contaminate the farming area, which would consequently affect the products from the farms and continually influence the livelihood of the community.

### 1.1. Oyster Import and Export Statistics

Asia was the largest oyster and shellfish exporter until 2004, but currently, Europe is the leading exporter in the world [1]. In Asian countries, the major exporters are South Korea, China, and Japan, even though Southeast Asian countries are also constant exporters of oysters. According to the Southeast Asian Fisheries Development Center, Southeast Asia's total production of oysters from aquaculture in 2016 was 40.5 thousand tonnes, which was mainly produced in Thailand and the Philippines [2]. Compared to other export goods from Thailand, bivalves including oysters still share a small portion of the market. Thailand exported $290k worth of oysters in 2016, and imported $3.8 million in the same year. The export value from 2015 to 2016 dropped by more than 50% of the amount exported before 2014. In parallel, the import value gradually increased during the past decade and reached a maximum of $3.8 million in 2016. Oyster consumption in Thailand may increase due to the higher domestic demands, including the expanding of the catering industry. The revenue of the tourism industry is shifting toward high-end tourism, which requires fresh and high-quality seafood to serve in the fine-dining restaurants. Besides, domestic oyster consumption is gaining more popularity than in the last decade due to its rich nutrition at affordable prices [3].

### 1.2. Oyster Culture in Thailand

Thailand, known as a country rich in oysters and bivalve products for 50 years, is facing a decrement in these natural bivalve products due to the overconsumption and unlimited harvest from the gradual expansion of the coastal community. The decline in natural harvesting with increasing demand from the domestic consumption market encourages the government to boost coastal aquaculture development. Three main species involved in aquaculture are shrimp, fish, and bivalves. Most of the area is devoted to shrimp cultivation due to it having the highest economic value compared to the total export value of coastal farming. Oysters are sensitive to environmental conditions, and other than the brackish water requirement, its survival rate and the density of oysters per cultivating area are largely influenced by alkalinity, salinity, pH, $NO_3$, and $NH_4$-N of the water [4]. Thus, only the coastal area with suitable conditions is suited for cultivation. Through the Department of Fisheries, education and financial support were provided to the project initiative area, Chonburi, Rayong, and Chantaburi Province [5].

Aquaculture farming later spread to the southern and eastern parts of Thailand. Nowadays, the two most well-known oyster-culture areas are Ban Don, Suratthani Province, and Ang Sila, Chonburi Province. The applied techniques include both intensive aquaculture and extensive aquaculture; in some areas, the semi-intensive aquaculture is developed to avoid the abrupt environmental conditional changes and optimize the oysters' larvae attachment and growth rate. Crassostrea belcheri, Saccostrea cucullata, and Crassostrea iredalei are the three main species widely cultivated for commercial demand [3]. Apart from the oyster, blood cockle and green mussel are two other popular cultivations done at the same time. The oyster yield from coastal aquaculture was 14.4 thousand tonnes in 2016, which is approximately 10% of the total shellfish culture (Figure 1), while blood cockle shares 31%, and the majority goes to green mussels, which take 59% of the total cultivation [6].

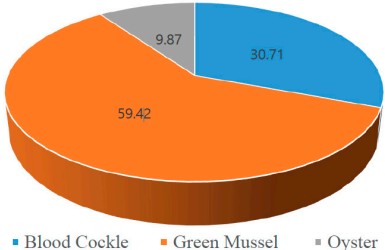

**Figure 1.** Thailand's total bivalve cultivation from aquaculture in 2016.

## 1.3. Oyster Culture in the Philippines

The fisheries sector in the Philippines is important for the economy to provide sustainable employment and income, and there has been a positive growth rate in fisheries production since the year 2000 [7]. In particular, the Philippines is an archipelago that has numerous suitable sites for oyster farming. The 17 provinces of oyster farms are located in the regions I, IV, and VI. In particular, Cavite and Pangasinan account for the majority of the oyster production [8].

The Bulacan and Cavite regions in the Philippines, which are close to Metro Manila City, capital of the country, accounted for approximately 28% of the total national oyster production (5.9 million tons per year) during the period of 2015–2017 [8]. Raw oysters and byproducts comprise a major part of the local fishery industry. As oyster and mussel farming are growing industries with little capital investment and moderate labor skills in the Philippines, the livelihood of the local communities heavily relies on these industries; currently, the Philippines has larger potential industries, suggested by FAO for the Philippines Government [9].

In the Bulacan region, oyster farming is highly vulnerable due to urbanization and global warming. A lack of adequate waste treatment has caused serious problems by the outlet of oyster waste. Marilao River in Bulacan particularly carries untreated waste to Manila Bay and the residents nearby Marilao are still dumping their garbage into the river. Moreover, a rise in mean air temperature can stimulate the occurrence of red tide. Although Bulacan has been free from red tide, there is still a possibility of the occurrence of red tide due to discharge from waste and climate change in the future [10].

Cavite is another place where oyster farming is a major industry that has a similar problem to Bulacan. The Cavite local government warned local government units (LGUs) that sanctions and punishment will be implemented in the case of not shutting down open dump sites, since open dump sites are illegal under the law. More concern comes from the fact that Cavite, where more than 3.6 million residents are living, does not have a landfill area [11].

## 1.4. Climate Changes on Bivalve Farming and the Shell Waste Problems

The Bulacan and Cavite regions are facing serious environmental pollution with oyster farming of the local communities and the side effects of climate change. Discharging eutrophic waste water from the urban areas and an increase in air temperature generate a possibility of red tides alongside the coastal areas. This harmful algal bloom is catastrophic to oyster farms, contaminating and killing oysters [12]. Furthermore, ocean acidification, caused by the increase in the atmospheric $CO_2$ concentration, also suppresses oyster production by accelerating the decomposition of the oyster shell. These environmental disasters can deteriorate the environmental quality and livelihood of the local community.

However, with consideration of the substantial production of oysters in Bulacan and Cavite, the waste oyster shell also needs better management. Occurrences of disease and noxious odor are particularly concerning in the cases of illegal dumping in various places and open storage of waste oyster shells. According to the standard cost notification by the Environmental Ministry, The Republic of Korea, the landfill disposal cost (digging, garbage selection, and transfer to the neighboring sanitary landfill) is approximately US$50 per ton. Given there are multiple side effects of dumping oyster shell waste from the perspective of environmental, hygienic, social, and financial dimensions, we need to pay more attention to the oyster shell recycling from the beginning state of the oyster production.

The estimated amount of waste from oysters and other shellfish is not available in detail. However, the oyster shell waste is categorized as Municipal Solid Waste (MSW) or commonly considered as trash or garbage. Although the waste shell is mostly from the local industry and restaurant, a bivalve shell is classified as food waste from the household. The generated household waste was nearly 1.13 kg per each person in one day as reported by Thailand's state of pollution report in 2017. The total amount of waste generated was 27.37 million tons, and in contrast, only 8.51 million tons (31%) were utilized, leaving 18.86 million tons as accumulated waste, where 11.69 million tons were properly disposed of in the designated place and 7.17 million tons were improperly disposed of (Figure 2) [13].

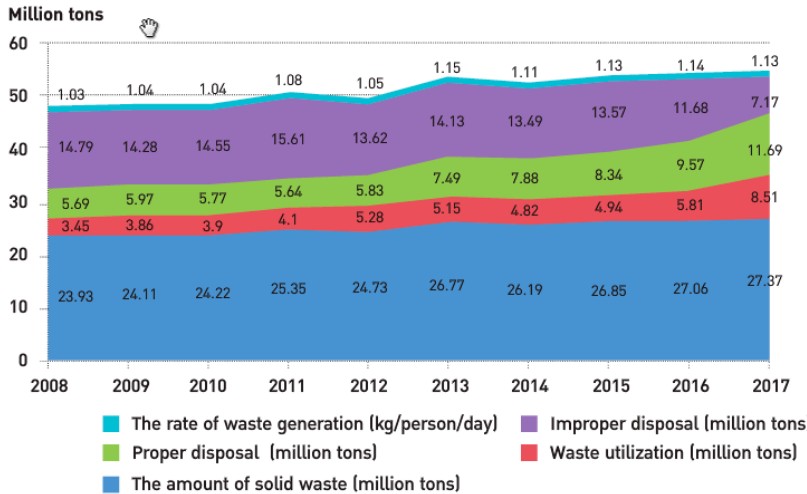

**Figure 2.** The municipal solid waste statistics in Thailand from 2008 to 2017 [13].

Pollution and global climate change, thus, are environmental and economic issues, substantially affecting the local communities. Furthermore, as this industry has a strong potential for export, these issues need to be solved in the case of quality assurance. Consequently, improving the environmental condition of the coastal regions for oysters is a timely issue in these regions. In this context, we proposed a possible project for the solution to these issues.

*1.5. Current Utilization of Waste Oyster Shells*

The oyster shell waste reutilization methods are not comprehensive knowledge in Thailand; most of the shell use in Thailand is still in the small-scale raw application of the waste shell. The villagers in the oyster farming area use the ground up, dried shell to feed the geese since the shell is a source rich in calcium. The small marine organisms and larvae use the artificial reef of oyster shells for their shelter; this oyster-shell reef not only attracts marine life but also helps mitigate some of the coastal erosion. Kuykendall et al. [14] suggest the use of cracked oyster shell as clutch materials for oyster recruitment. The cracked oyster shells have a better ability in oyster recruitment than the whole shells. Nevertheless, the application of the waste shell is not only limited to the raw usages.

The study by Klathae [15] described the utilization of oyster waste as the component for interlocking block production. The different ratio of crushed oyster shell powder from 0 to 50 percent was used as cement substitution. The study showed that blocks with 10 and 20 percent of crushed oyster shell have the comparable pressure resistance of the commercial interlocking blocks. The similar methods were acknowledged in various areas, especially in the aquaculture zone. In many oyster farming villages, the waste oyster shell is burnt and applied in cement-substitute manufacture. The oyster shell's main component is $CaCO_3$ which transforms into lime, CaO, under the proper heat treatment. The produced CaO can be used as a mixture for Portland cement production and its application in construction, agriculture, and painting.

Oyster shell also has some application in the water reservoir system. Sawawin et al. [16] described the study of using the lime produced from the waste shell for water treatment by the Alum Coagulation Process. Lime utilized from waste shell have the comparable capability in pH adjustment and coagulation media to commercial lime. The researcher noticed that the aggregation and sedimentation of the colloid particles were faster and the turbidity of the wastewater reduced significantly with the application of shell lime. A similar adaptation of waste oyster shells in Thailand is as filter media for the fish pond system. Oyster shells have an exceptionally stabilizing effect on the pH value, an indicator for water carbonate hardness, and consequently, indirectly on the pH value. However, this utilization of waste shell is still on a small scale, none of the processes advanced enough to be commercialized or applied in the country's waste processor units.

The main objectives in this study were to characterize the mollusk shells and natural limestone by XRD and XRF analysis, and also synthesize the aragonite needles from both limestone and different mollusks shells by the carbonation process. The optimum experimental conditions are explained and discussed in the methodological part for preparing the needle-like aragonite $CaCO_3$.

## 2. Materials and Methods

We developed a method for the synthesis of aragonite needles from different mollusk shells and natural limestone by the carbonation process. In this process, we collected different mollusk shells from the coastal region of Korea and cleaned them with water and ethyl alcohol for removing impurities which were attached to the shells. These shells were calcinated at 1000 °C for 120 min by using an electric furnace. Figure 3 shows the raw mollusk shells with natural limestone and calcinated limestone and different calcinated shells. These calcinated samples were mechanically ground for 60 min until the particle size was <100 μm. The fine powder was used for the synthesis of aragonite needles by the carbonation process.

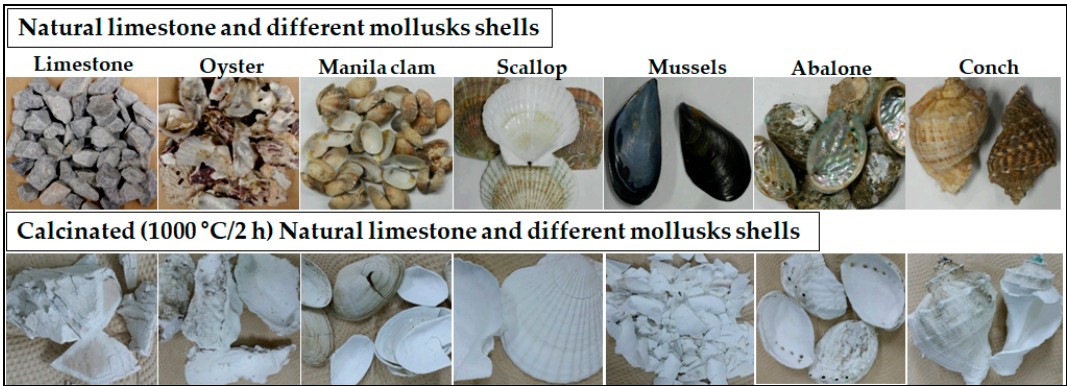

**Figure 3.** Raw mollusk shells with limestone and calcinated mollusk shells with calcinated limestone at 1000 °C/ 2 h.

*Aragonite Synthesis from Limestone and Different Mollusks Shells by Carbonation Process*

The aragonite needles were synthesized from limestone and mollusk shells by the carbonation process. In this process, we used calcinated limestone and mollusk shell powders individually. The calcinated and mechanically ground mollusk shell fine powder (<100 μm) was washed (100 g/lit) three times with distilled water at room temperature by the hydration process; after the filtering process, the samples were dried at 80 °C for a 12 h duration.

The dried calcium ($Ca^{2+}$)-rich powders of limestone and mollusk shells were used for the synthesis of aragonite needles by the carbonation process. In this process, we used 26 g of each mollusks shell and limestone calcium-rich dried samples were added individually to 0.6 M of $MgCl_2$ solution, and $CO_2$ gas was injected (50 mL/min flow rate) into the $MgCl_2$-$Ca^{2+}$-rich mollusk shell powder suspension at pH 11 for aragonite form of needle synthesis by carbonation reaction at 80 °C for 3 h duration time, as described in Figure 4 [17]. In the agaronite form of $CaCO_3$ synthesis process, the $Ca^{2+}$ ions from mollusk shells reacted with hydrated carbon dioxide ($CO_3^-$) at 80 °C for 3 h reaction time. The different mollusk shells and limestone samples were used individually for the carbonation process in the same experimental conditions for the synthesis of aragonite needles.

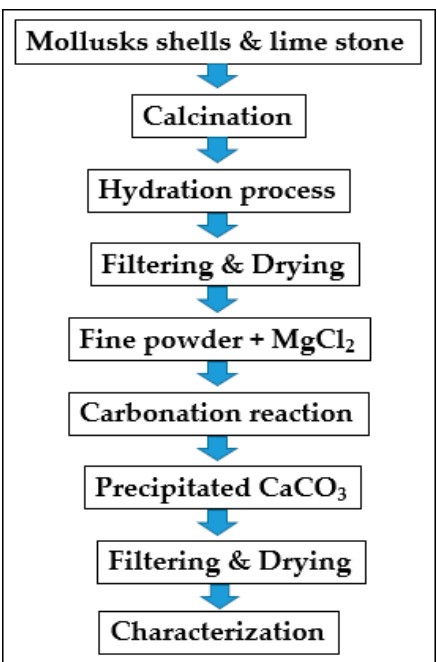

**Figure 4.** Aragonite needles synthesis from mollusk shell and limestone by carbonation process.

## 3. Results and Discussion

*Chemical Composition of Raw and Calcinated Samples of Mollusk Shells and Limestone*

The major chemical composition of mollusk shells and natural limestone is $CaCO_3$ (98 to 99 wt%). Figure 5 shows the XRD patterns of different waste raw mollusk shells with natural limestone. The results indicated that the raw oyster shells and limestone major phases were $CaCO_3$, identified by XRD patterns. The natural limestone had 95.49 wt% of calcite and 1.78 wt% of aragonite; the shells of the mollusks such as oysters had 92.66 wt% of calcite and 2.71 wt% of aragonite; manila clam shells had 6.15 wt% of calcite and 92.67 wt% aragonite; scallop shells had 94.17 wt% of calcite and 3.24 wt% aragonite; mussels shells had 72.04 wt% of calcite and 24.78 wt% of aragonite; abalone shells had 22.19 wt% calcite and 75.95 wt% aragonite; and conch shells had 59.28 wt% of calcite and 37.93 wt% of aragonite form of $CaCO_3$. These results indicated that mollusk shells and natural limestone chemical compositions are similar. Due to this reason, instead of natural limestone, mollusk shells recycling is most useful for industrial applications and also beneficial for reducing the waste from coastal regions.

The XRF results in Table 1 illustrate the mollusk shell and limestone chemical composition was mostly CaO (53 to 55 wt%) and there was ignition loss (44 to 46 wt%) as $CO_2$ gas was released from the $CaCO_3$ decomposition. The results show the chemical compositions of natural limestone and mollusk shells are almost similar and the waste shell recycling and reutilizing is more beneficial for industrial applications.

**Table 1.** XRF analysis of the different mollusk shells with natural limestone.

| Composition (Wt%) | $SiO_2$ | $Al_2O_3$ | $Fe_2O_3$ | CaO | MgO | $K_2O$ | $Na_2O$ | $TiO_2$ | MnO | $P_2O_5$ | Ig Loss |
|---|---|---|---|---|---|---|---|---|---|---|---|
| Limestone | 0.11 | 0.03 | 0.09 | 55.54 | 0.20 | 0.03 | <0.02 | <0.01 | 0.1 | 0.01 | 43.79 |
| Oyster | 0.45 | 0.12 | 0.06 | 53.66 | 0.26 | 0.06 | 0.55 | <0.01 | 0.01 | 0.16 | 44.56 |
| Manila | 0.12 | 0.03 | 0.02 | 54.27 | <0.01 | 0.03 | 0.65 | <0.01 | <0.01 | 0.06 | 45.06 |
| Scallop | 0.07 | 0.01 | 0.02 | 54.96 | 0.05 | 0.03 | 0.42 | <0.01 | <0.01 | 0.12 | 44.09 |
| Mussels | 0.07 | 0.01 | <0.01 | 52.83 | 0.17 | 0.02 | 0.38 | <0.01 | <0.01 | 0.06 | 46.17 |
| Abalone | 0.15 | 0.04 | 0.01 | 53.04 | <0.01 | 0.03 | 0.59 | <0.01 | <0.01 | 0.02 | 46.23 |
| Conch | 0.11 | 0.02 | 0.01 | 54.59 | 0.09 | 0.03 | 0.53 | <0.01 | <0.01 | 0.02 | 44.68 |

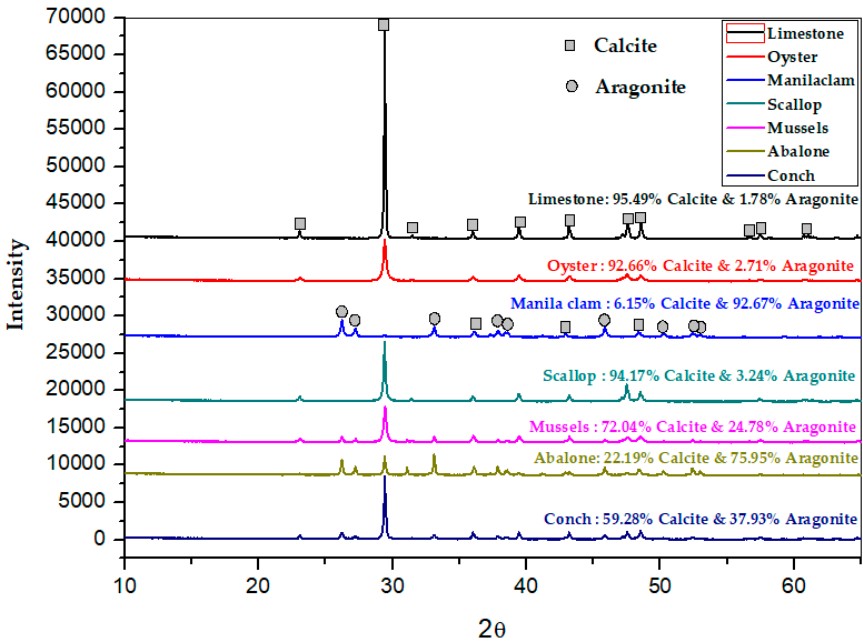

**Figure 5.** XRD patterns of raw mollusk shells with natural limestone.

The mollusks shells and natural limestone calcination process (1000 °C for 2 h) can produce pure lime (CaO); as confirmed by the results of XRD patterns, calcinated (1000 °C for 2 h) mollusk shells and limestone have major chemical compositions of lime (CaO) (98–99 wt%) as shown in Figure 6. Based on these experimental results, the similar chemical composition of lime (CaO) was generated from both mollusk shells and limestone; additionally, those samples exhibited similar chemical properties. These lime (CaO) samples utilized for industrial applications may also exhibit the same chemical properties.

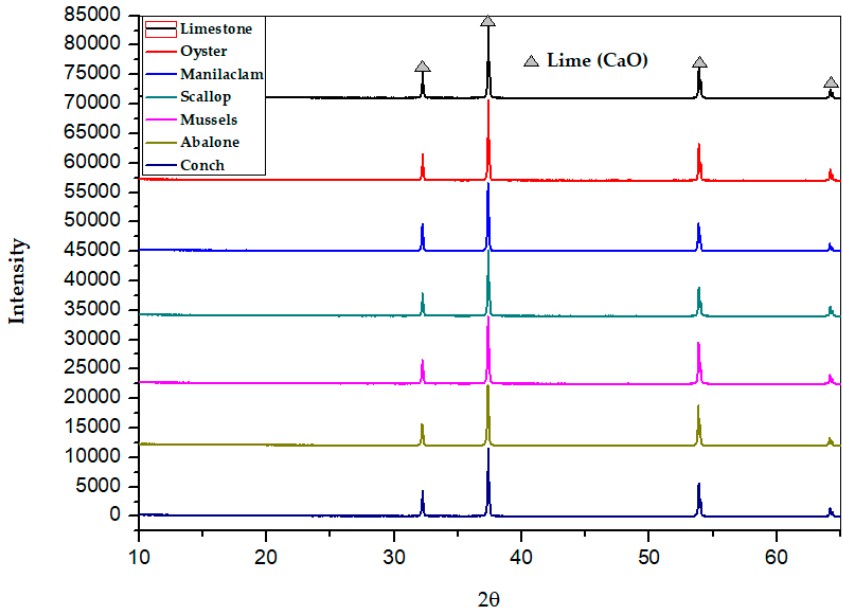

**Figure 6.** XRD patterns of calcinated (1000 °C/ 2 h) mollusk shells and limestone.

In the results shown in Figure 7, XRD patterns of pure aragonite needles were synthesized from purified limestone and different mollusk shell samples by the carbonation process at 80 °C for 3 h duration time. The results show that almost all different shell and limestone samples produced purified

aragonite needles. The overall experimental summary depicts that the chemical properties of limestone and oyster shells were similar and the recycling process of different mollusk shells is more beneficial for saving the natural limestone for future generations and to avoid the mollusk shells as a waste material.

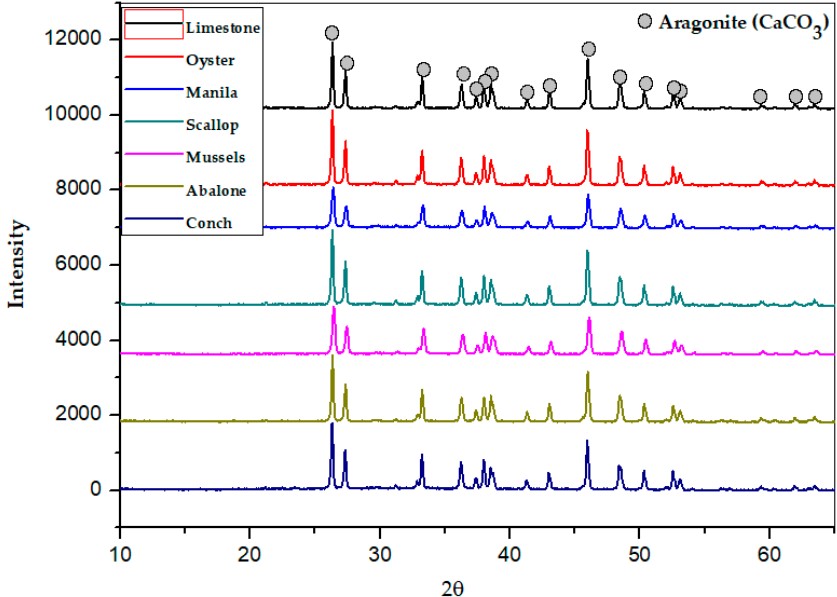

**Figure 7.** XRD patterns of aragonite precipitated $CaCO_3$ by the carbonation from different mollusk shells and limestone samples at 80 °C, 3 h duration time.

## 4. Conclusions

With the application of this technology, the cumulatively dumped oyster shells could be recycled into value-added products, and the environment of neighboring communities can expect to be improved. Treating waste oyster shell primarily solves hygienic issues, including awful smell, pest occurrence, and disease spread. Production of ecofriendly cement can also partially mitigate $CO_2$ emission, which can contribute to attaining national greenhouse gas mitigation goals with international support [18]. In addition, the provision of calcium oxide will mitigate sulfur dioxide releases from factories and thermal power plants, contributing to the quality of air in urban areas. Disastrous red tides and ocean acidification in Manila Bay and other oyster farming regions can be reduced by the application of calcium oxide technology to the oyster farms. This measure can reduce the vulnerability of the local livelihood by climate change. In addition, operation and possible scale-up of the treatment plant will generate more employment in the local communities, and also lead to research and development of new climate technology, ultimately to the establishment of a new industry.

The shells are rich in calcium carbonate, which comprises around 95% of their composition, and can be used for repairing damaged oyster reefs due to the red tides caused by the global climate change as explained above [19]. Ground and well-treated oyster shell powder can be of great help to neutralize the acidity of the coastal ocean, providing a better condition for oyster shells to grow into eco-friendly and sustainable products. Furthermore, the ecotourism industry based on a coral reef will be sustainable, preserving coral reef ecosystems.

The present study aimed to explore the viability of the recycling of mollusk shells from coastal regions and reducing its environmental toxicity. The experimental summary reveals the mollusk shell waste reutilization as an alternate material of natural limestone, because of the mollusks shells having a large amount of calcium source with less impurities. The recycling process of mollusk shells offers the wide application in various industries. The preparation of aragonite needles from mollusk shells by the carbonation process is also an economic method for industrial application due to the wide source of mollusks shells, and also the utilization of $CO_2$ gas by the carbonation process is beneficial to save

the atmospheric conditions. However, the utilization of mollusks shell waste is beneficial to reduce the environmental toxicity and save the natural resources such as limestone for the next generations.

**Author Contributions:** R.C., T.T., C.T. did the experiments and analyzed the data and wrote the manuscript. E.J.S. collected the literature. J.W.A. corrected the final manuscript and agreed to submit this data to the sustainability journal.

**Funding:** This research was funded by the National Strategic Project-Carbon Mineralization Flagship Center of the National Research Foundation of Korea (NRF) funded by the Ministry of Science and ICT (MSIT), the Ministry of Environment(ME) and the Ministry of Trade, Industry and Energy (MOTIE).(2017M3D8A2084752).

**Acknowledgments:** We are greatly thankful to our co-authors and corresponding author Ji Whan Ahn.

**Conflicts of Interest:** The authors declare no conflict of interest.

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
