# Peer review of "Sustainable Solutions for Oyster Shell Waste Recycling in Thailand and the Philippines"

_recycling, doi:10.3390/recycling4030035_

Round 1

Reviewer 1 Report

This paper is an important contribution to the management of mollusk shell waste  in high production areas.

The text is well organized and written clearly and sequentially.

Line 176 states "We have developed a technology for the synthesis of aragonite needles from different mollusk shells, ...". Why is the title of the paper "Sustainable Solution for Oyster Shell Waste in Thailand and the Philippines"?

The paper should emphasize the applications of aragonite needles to justify and highlight the importance of the study.

On lines 205 and 206 it states that "... the waste shells recycling for industrial applications is also economic as well as environmentally friendly." The process used consume energy and release CO2 so, this study, in a second phase should include a life cycle analysis that assesses the impacts of its use. In my opinion this should be referred to in the paper. An economical study could be also important.

Line 217 I think it's from instead of "form"

Line 238 Why is "... more beneficial for industrial applications instead of utilizing the natural limestone."?

Author Response

Reviewer - 1

This paper is an important contribution to the management of mollusk shell waste in high production areas.

The text is well organized and written clearly and sequentially.

Line 176 states "We have developed a technology for the synthesis of aragonite needles from different mollusk shells,”. Why is the title of the paper "Sustainable Solution for Oyster Shell Waste in Thailand and the Philippines"?

The paper should emphasize the applications of aragonite needles to justify and highlight the importance of the study.

Ans). We modified the title and this study can explain the oyster shell waste generated from Thailand and Philippines and recycling method for sustainable solution.

On lines 205 and 206 it states that "... the waste shells recycling for industrial applications is also economic as well as environmentally friendly." The process used consume energy and release CO2 so, this study, in a second phase should include a life cycle analysis that assesses the impacts of its use. In my opinion, this should be referred to in the paper. An economical study could be also important.

Ans). We modify the sentence in line 205-206.

Line 217 I think it's from instead of "form"

Ans). We changed the word “form” to from

Line 238 Why is "... more beneficial for industrial applications instead of utilizing the natural limestone."?

Ans). The chemical composition of limestone and oyster shell wastes almost similar. The utilization of oyster shell waste are the major advantage for saving the natural limestone for future generations and avoid the oyster shells as a waste material.

I modified the line 237-240, as per your suggestions.

Reviewer 2 Report

Please, look at the attached file for the review comments. 

Author Response

Reviewer -2.

Manuscript Title: Sustainable solution for oyster shell waste in Thailand and the

Philippines

1. The tile of the paper is interesting and the study is important from the viewpoint of

environment and sustainable development.

Ans). Yes sir, thank you

2. In the introduction part, there are two things missing: first, it should mention the outline orstructure of the paper and also state a research question or hypothesis with purposestatement.

Ans). We arranged proper way of the outlines of the paper, and our research objectives also clearly mention in line 175-178.

3. Inclusion of a conceptual framework demonstrating the oyster shell footprint or flowmechanism and processing or recycling or reusing would clarify the environmental and

economic concerns of the related issues.

Ans).We explained the recycling of mollusks shells and environmental benefits with economic values was explained in conclusions part in line 272-277.

4. The organization of the manuscript is not as per the structure of a scientific paper. Thereshould be a separate section for Materials and Methods in which how the oyster wastehad been collected, recycled/reused and processed with clear mechanical and chemicalprocess and reaction equations/mechanisms.

Ans).We separated the materials and methods as per your suggestions, in sub section -2 (line 179).

5. There should be separate section of Results and Findings in which all data from XRDand XRF should presented, described and interpreted. The methods and results sectionshould be separated and described/analyzed and interpreted in their respective sections.

Ans).We separated results part in separate sub section -3, Methods part also separated as per your suggestions.

6. There should a section for Implications of Findings in terms of Environmental

Implications and Economic Implications , for example.

Ans).Weexplained the environmental implications and economic benefits were explained in conclusions part in line 272-277.

7. The English writing should be improved massively.

Ans).We gave English corrections for improving sentences.

I hope authors will follow these advice so that the paper will be more organized and better inlanguage.

Good Luck !

Thank you sir

Round 2

Reviewer 2 Report

Certainly, the manuscript has been improved from the earlier version. However, there are many language related issues to be addressed by the authors. For example: 

Line 21, the highlighted part have also discussed does not seem correct grammatically. I think it should be have also been discussed. 

Line 29, the brick water does not make sense. I think it should be the brisk water

Line 31, in generally should be either in general or generally (without in). 

Lines 32-34, the sentence ‘Low fat, calories, …… consumers.’ does not have a proper verb and the sentence seems incomplete. 

The list may continue… instead, I have highlighted the text in the manuscript with my serious concerns with the language part. The manuscript needs to be edited again for English. As a reviewer, I can only indicate it but cannot do a massive editing (sorry for that).  Please, look at the highlighted part in the attached copy of the manuscript, but the editing for English should not be limited only to these parts, rather should be done thoroughly.  

Also, I thought but forgot to mention in my earlier review, that experimental design is not detailed enough. Please, make it as detailed as possible so that anybody who wants to replicate the experiment should be able to do it to get similar results. 

In implications, please explain, in what way this experimental design is different from Ramakrishna et al. (2016) so that it helps scientists to understand the process (in terms of new or the same outcomes). 

Author Response

Comments and Suggestions for Authors

Certainly, the manuscript has been improved from the earlier version. However, there are many language related issues to be addressed by the authors. For example: 

Line 21, the highlighted part have also discussed does not seem correct grammatically. I think it should be have also been discussed. 

Ans). We changed line 21, as per your suggestion.

Line 29, the brick water does not make sense. I think it should be the brisk water

Ans). We changed the word “brick” to “brisk” in line 29.

Line 31, in generally should be either in general or generally (without in). 

Ans). We changed the word “in general” as per your suggestion in line 31.

Lines 32-34, the sentence ‘Low fat, calories, …… consumers.’ does not have a proper verb and the sentence seems incomplete. 

Ans). We verify the entire manuscript as per your suggestion we modify the sentences and avoid the grammar mistakes

The list may continue… instead, I have highlighted the text in the manuscript with my serious concerns with the language part. The manuscript needs to be edited again for English. As a reviewer, I can only indicate it but cannot do a massive editing (sorry for that).  Please, look at the highlighted part in the attached copy of the manuscript, but the editing for English should not be limited only to these parts, rather should be done thoroughly.  

Also, I thought but forgot to mention in my earlier review, that experimental design is not detailed enough. Please, make it as detailed as possible so that anybody who wants to replicate the experiment should be able to do it to get similar results. 

In implications, please explain, in what way this experimental design is different from Ramakrishna et al. (2016) so that it helps scientists to understand the process (in terms of new or the same outcomes). 

Ans). We modify the materials and methods part as per your suggestion and explain the experimental procedure in figure 4, clearly.